# Provable Accuracy Collapse in Embedding-Based Representations under Dimensionality Mismatch

Dionysis Arvanitakis [1]   Vaggos Chatziafratis [2]   Yiyuan Luo [2]

## Abstract

Embedding-based representations in Euclidean space $\mathbb{R}^d$ are a cornerstone of modern machine learning, where a major goal is to use the *smallest dimension* that faithfully captures data relations. In this work, we prove sharp dimension–accuracy tradeoffs and identify a fundamental information-theoretic limitation: unless the embedding dimension $d$ is chosen close to the ground-truth dimension $D$, accuracy undergoes a sudden collapse. Our main result shows that this phenomenon arises even in standard contrastive learning settings, where supervision is limited to a set of $m$ anchor–positive–negative triplets $(i, j, k)$ encoding distance comparisons $\mathrm{dist}(i,j) < \mathrm{dist}(i,k)$. Specifically, given triplets realizable by an unknown ground-truth embedding in $D$ dimensions, we prove that there exists constant $c < 1$, such that *every embedding of dimension at most $cD$ violates almost half of the triplets*, yielding accuracy as low as a trivial one-dimensional solution that ignores the input. We complement our information-theoretic bounds with strong computational hardness results: under the Unique Games Conjecture, even if the given triplets are nearly realizable in $D = 1$ dimension, no polynomial-time algorithm—*regardless of its dimension*—can achieve accuracy above the trivial $50\%$ baseline.

## 1. Introduction

Embedding-based representations, particularly in Euclidean space $\mathbb{R}^d$, have become a central component of modern machine learning pipelines (Mikolov et al., 2013a;b; Bengio et al., 2013; LeCun et al., 2015). By mapping data into $d$-dimensional vectors while ensuring that semantically similar items remain close in the embedding space, such representations enable a unified approach to semantic search and reasoning across diverse modalities, including graphs, text, audio, images, and code (Grover & Leskovec, 2016; He et al., 2016; Chen et al., 2020).

A key design parameter in embedding-based approaches is the *representation dimensionality* $d$, with common choices ranging from a few hundred to a few thousand dimensions (Radford et al., 2018; Comanici et al., 2025). For example, modern foundation models often maintain very high-dimensional latent spaces (e.g., 3072), while downstream embeddings are typically projected into lower-dimensional vectors (e.g., 128) for efficiency and scalability, or are fine-tuned depending on the task at hand (Kusupati et al., 2022).

In this work, we aim to highlight the critical role of embedding dimensionality by characterizing fundamental limitations on how the choice of $d$ impacts model accuracy. On the one hand, higher-dimensional embeddings offer greater expressive power, enabling finer-grained distinctions between data points. On the other hand, increasing dimensionality incurs substantial computational and operational costs: memory usage, storage requirements, and search latency all grow with $d$, while inference becomes slower and more energy-intensive. Unfortunately, despite the widespread use of embedding-based models and explicit efforts to fine-tune dimension (Kusupati et al., 2022), it remains unclear how accuracy (on training or downstream tasks) is impacted by the dimension $d$ and what the fundamental limitations are.

For example, if a model's dimension is increased, say from 256 up to 512, should we expect to see a significant improvement in performance? Recent empirical works on dimension truncation consistently observe across 6 state-of-the-art text encoders and 26 downstream tasks Takeshita et al. (2025); Tsukagoshi & Sasano (2025), that while randomly removing up to 50% of embedding dimensions results in only a minor drop in performance (less than 10%), beyond a certain truncation threshold ($\sim 90\%$), *accuracy drops quite rapidly*.

### 1.1. Our Setting: Contrastive Triplet Embeddings

Our starting point for our work is to investigate the aforementioned rapid deterioration of model accuracy, when-

---

[1]Northwestern University, IL, USA [2]University of California, Santa Cruz, CA, USA. Correspondence to: Dionysis Arvanitakis <dionarva@u.northwestern.edu>.

*Proceedings of the 43$^{rd}$ International Conference on Machine Learning*, Seoul, South Korea. PMLR 306, 2026. Copyright 2026 by the author(s).

ever the embedding dimension crosses a certain threshold, as it currently lacks theoretical grounding for why and when it might occur. Towards formally understanding this behavior and to estimate the threshold, we focus on a standard contrastive learning setting, based on the popular anchor-positive-negative paradigm used in the Triplet Loss and NCE Loss (Gutmann & Hyvärinen, 2012; Schroff et al., 2015; Saunshi et al., 2019; Vankadara et al., 2023; Avdiukhin et al., 2024; Alon et al., 2024).

In this context, we are given a collection $\mathcal{T}$ of $m$ *triplet comparisons* of the form "item $i$ is more similar to $j$ than to $k$," indicating that distances in the final embedding should satisfy $\mathrm{dist}(i, j) < \mathrm{dist}(i, k)$.[1] The accuracy measure here corresponds to the probability that the embedding-induced ranking agrees with such pairwise relevance constraints, or equivalently, we want to find embeddings that agree with as many of the triplets as possible:

**Definition 1.1** (Accuracy). Let $V = \{1, \ldots, n\}$ be a set of items and $\mathcal{T} \subseteq V \times V \times V$, be a set of $m$ triplets $(i, j, k)$. Given an embedding $f : V \to \mathbb{R}^d$, we say that a triplet $(i, j, k)$ is *satisfied* if $\|f(i) - f(j)\|_2 < \|f(i) - f(k)\|_2$ and *violated* otherwise. The *accuracy* of the embedding is defined as the fraction of satisfied triplets in $\mathcal{T}$ ($|\mathcal{T}| = m$):

$$\mathrm{acc}(f; \mathcal{T}) = \frac{1}{m} \sum_{(i,j,k) \in \mathcal{T}} \mathbf{1}[\|f(i) - f(j)\|_2 < \|f(i) - f(k)\|_2]$$

The main driving question behind our work is:

*Given $m$ triplets of the form "$i$ is closer to $j$ than to $k$," how does the embedding dimension $d$ affect the accuracy?*

There are three main reasons why we focus on such contrastive learning tasks, where we must preserve distance *comparisons*, rather than distance *lengths* as in a metric embedding (Bourgain, 1985; Linial et al., 1995; Dasgupta & Gupta, 2003; Indyk, 2001; Larsen & Nelson, 2017). First, as we will see, our main results provide information-theoretic and computational *limitations* of embedding-based representations, so focusing on "easier" embedding tasks makes our lower bounds stronger. Second, contrastive triplet tasks are ubiquitous in metric learning and ordinal embeddings (Bilu & Linial, 2005; Alon et al., 2008; Vankadara et al., 2023; Chatziafratis & Indyk, 2024; Alon et al., 2024; Avdiukhin et al., 2024), as they capture nearest-neighbor applications and various retrieval tasks. For example, in document retrieval, a triplet $(i, j, k)$ specifies that for query $i$, item $j$ should be ranked ahead of item $k$. Third, empirical works by Gutmann & Hyvärinen (2012); Schroff et al. (2015); Saunshi et al. (2019; 2022) have established that models trained with various continuous/smooth proxy objectives for

---

[1]We use $(i, j, k)$ or the more common notation $(i, j^+, k^-)$ to specify this distance comparison.

triplet accuracy achieve high performance on downstream tasks, so shedding light on dimension-accuracy tradeoffs is well-motivated.

## 1.2. Our Contributions

As we vary the dimension $d$, we care both about *information-theoretic* limitations in triplet embeddings, i.e., what is the best achievable accuracy by *any* model of dimension $d$, independently of the optimization method or loss function or architecture used, and about *computational* limitations, i.e., can we efficiently find an embedding with high accuracy.

**Trivial $50\%$-Baseline and Accuracy Collapse.** Observe that finding an embedding with $50\%$ accuracy is always trivially achievable, even using one-dimension:

**Definition 1.2** (Trivial $50\%$-Baseline). Let $f : V \to \mathbb{R}$ be a one-dimensional embedding, where $f(i)$ is sampled i.i.d. from a continuous distribution on $\mathbb{R}$. In particular, $f$ is independent of the given triplet constraints.

Such an embedding entirely ignores the given triplets, yet it achieves $50\%$ accuracy: for a fixed $(i, j, k)$, symmetry implies that $\mathbb{P}(|f(i) - f(j)| < |f(i) - f(k)|) = \frac{1}{2}$. An important concept for our work is that of *accuracy collapse*, capturing the scenario where no $d$-dimensional embedding can satisfy more triplets than this trivial $50\%$-baseline.

**Information-Theoretic Limitations.** Our first result helps explain empirically-observed sharp drops in accuracy (Takeshita et al., 2025; Tsukagoshi & Sasano, 2025), via an information-theoretic lower bound for triplet embeddings under dimension mismatch up to a *constant* factor:

**Theorem 1.3** (Dimension-induced accuracy collapse for realizable triplets). *For every integer $D \geq 240$, and for every $\varepsilon > 0$, there exists a collection of triplet constraints $\mathcal{T} \subseteq V \times V \times V$ with the following properties:*

1. *(Realizability) There exists an embedding $f^\star : V \to \mathbb{R}^D$ that satisfies all triplets in $\mathcal{T}$.*

2. *(Accuracy Collapse) There exists constant $c = c(\varepsilon) < 1$, such that for any embedding $f : V \to \mathbb{R}^d$ with $d \leq c \cdot D$, the fraction of satisfied triplets is at most*

$$\mathrm{acc}(f; \mathcal{T}) \leq \frac{1}{2} + \varepsilon$$

Interestingly, our result corroborates recent empirical works (Takeshita et al., 2025; Tsukagoshi & Sasano, 2025) where aggressive dimensionality truncation (roughly $\sim$ $90\%$) leads to severe accuracy drop, as it positions the critical threshold within a *constant factor* of the dimension $D$ of the fully-expressive model. This implies that for certain tasks, even a constant approximation to the ground-truth

dimension $D$ does not yield any better accuracy guarantees than the trivial 1-dimensional embedding. Another consequence of our result is that there exist certain tasks, for which augmenting the dimension, say from 256 to 512, will not lead to any measurable improvement, and this is independent of the optimization methods used.

In fact, our result holds even if we allow the ground-truth[2] dimension $D$ to grow with the instance size $|V| = n$, as long as $D = o(\sqrt{n})$. Moreover, our lower bounds can be extended to the case of *quadruplet* comparisons studied in ordinal embeddings (Bilu & Linial, 2005; Alon et al., 2008; Vankadara et al., 2023), where $(i, j, k, l)$ indicates that $\|f(i) - f(j)\|_2 < \|f(k) - f(l)\|_2$, even for $D = o(n)$.

The main takeaway from our information-theoretic lower bound is that embedding dimension acts as a sharp bottleneck, independently of the optimization method, loss function or model architecture: *below a certain constant fraction of the ground-truth dimension $D$, embeddings suffer from accuracy collapse, even on realizable instances.*

**Computational Hardness of Approximation.** We also study *non-realizable* instances, where triplet comparisons may contain errors (agnostic setting) and hence there is no embedding (in any dimension) that satisfies all triplets. In our previous result, the realizability assumption isolates representational limitations imposed by dimension alone from issues related to computational constraints, noise, optimization method used, or various other model misspecifications.

Since it is NP-complete to check realizability of a given set $\mathcal{T}$ of $m$ triplets even for $D = 1$ (Opatrny, 1979; Fan et al., 2020; Avdiukhin et al., 2024), the optimization goal becomes to find an embedding that approximates the accuracy of an optimal embedding, i.e., given a target dimension $d$, the goal is to find an embedding $f : V \to \mathbb{R}^d$ that *maximizes* the fraction of satisfied triplets. Surprisingly, there is currently no better approximation algorithm than the trivial 50%-baseline in 1-dimension, even if the algorithm is allowed to use higher dimensions $d > 1$. Our second result settles the approximability of the problem:

**Theorem 1.4** (Computational Hardness of Approximation)**.**
*Assuming the Unique Games Conjecture, for every $\varepsilon > 0$, it is NP-hard to distinguish between triplet instances $\mathcal{T}$ that admit an embedding (in any dimension) satisfying at least a $(1 - \varepsilon)$-fraction of the triplets and instances for which no embedding satisfies more than a $(\frac{1}{2} + \varepsilon)$-fraction of the triplets.*

In other words, no polynomial-time algorithm can guarantee

---

[2]For realizable instances, we refer to any embedding with perfect accuracy $\mathrm{acc}(f^\star; \mathcal{T}) = 1$ as the *ground-truth* (or fully expressive) embedding, and to the smallest such dimension $D$ as the *ground-truth dimension* of the instance.

accuracy exceeding $\frac{1}{2} + \varepsilon$ on triplet embedding instances, even when there exists an embedding achieving accuracy at least $1 - \varepsilon$. We emphasize that this hardness of approximation holds even for near-realizable instances in $D = 1$, and is *independent of the dimensionality* used by the algorithm.

### 1.3. Further Related Work

Perhaps a first attempt towards dimensionality reduction would be to apply standard metric embedding tools, such as the Johnson-Lindenstrauss lemma that yields dimension $d = O(\log n/\varepsilon^2)$ (Dasgupta & Gupta, 2003). However, due to the inevitable $(1 \pm \varepsilon)$-distortion, it is well-known that it fails for ordinal embedding settings, where we care to preserve rankings of distances, as they may flip almost all triplet (or quadruplet) comparisons (Alon et al., 2008).

Prior to our work, theoretical findings on ordinal embeddings concerned special cases (Bilu & Linial, 2005; Fan et al., 2020; Chatziafratis & Indyk, 2024; Avdiukhin et al., 2024; Alon et al., 2024), and to the best of our knowledge we are the first to provide tight characterizations for dimension-vs-accuracy proving the accuracy collapse phenomenon down to the 50%-baseline. Specifically, prior works by Bilu & Linial (2005); Chatziafratis & Indyk (2024); Avdiukhin et al. (2024) studied realizable instances, and showed that to preserve all triplet (or quadruplet) comparisons, dimension $d = O(\min\{n - 1, \sqrt{m}\})$ is always sufficient, and $d = \frac{n}{2}$ or $d = \Omega(\sqrt{m})$ may be needed in the worst-case. Note that in these works $d$ is prohibitively large for practical considerations. In contrast, we focus on the general case where we aim to obtain good accuracy relative to some user-specified ground-truth dimension $D$. Other positive results by Fan et al. (2020) study dense instances with $m = \Omega(n^3)$ triplets on the line ($D = 1$) and provide a PTAS, while Alon et al. (2024) study PAC-learnability/sampling complexity for small generalization error. Recent empirical works compare different ordinal embedding methods (Vankadara et al., 2023), or highlight limitations in retrieval applications (Weller et al., 2025).

Regarding approximation and computational complexity, the inability to beat the trivial random baseline is an intriguing phenomenon in the theory of approximation algorithms, formalized as *approximation resistance* by Håstad's celebrated work (Håstad, 2001). Notably, problems such as MAX-3SAT and other Constraint Satisfaction Problems (CSPs) are approximation resistant (Hast, 2005; Guruswami et al., 2008). For example, under Khot's Unique Games Conjecture (Khot, 2002)—a central open problem in complexity and hardness of approximation—all ranking CSPs and tree reconstruction CSPs are approximation resistant (Guruswami et al., 2011; Chatziafratis & Makarychev, 2023). Characterizing which problems are approximation resistant is currently an active research area (see, e.g., recent work-

shops at Daghstuhl (Bulatov et al., 2016; Grohe et al., 2018; 2022; Bodirsky et al., 2025)) and our work points to a new geometric CSP with this property.

## 2. Accuracy Collapse in Triplet Embeddings

Before proceeding with the technical details of the proofs for Theorem 1.3, we begin by a high-level overview of the necessary intermediate steps.

### 2.1. Proof Strategy

As above, let $\mathcal{T}$ be a collection of triplet comparisons on the $n$ items of $V$, and let $m$ denote the number of triplets in $\mathcal{T}$. In order to show Theorem 1.3, on the one hand, the instance must be realizable in dimension $D$, and on the other hand, it should be very far from realizable for any dimension $d$ that is a constant approximation of $D$, for a sufficiently small constant. Towards this we study random instances below.

**Average-Case Instances.** We use the probabilistic method (Alon & Spencer, 2016) to construct a suitable *random triplet instance* with the appropriate density, and then show that it has *both* properties with positive probability; this yields the conclusion that there exist realizable instances where accuracy collapse takes place, even if the dimension $d$ is a constant approximation to the ground-truth dimension $D$. A similar construction can be extended for general ordinal embeddings where comparisons on 4 items $(i, j, k, l)$ are also allowed. More precisely, for every $D \in \mathbb{N}$ larger than a constant (240 suffices), we sample uniformly at random $m = \Theta(Dn)$ triplets, by first choosing at random distinct $x, y, z \in V$, and then choosing one of the possible triplets on them at random. We say that an instance $\mathcal{T}$ is sampled according to $\mathcal{I}(n, m)$ in that case. In the following theorem, we show that a random instance with $\Theta(Dn)$ constraints is satisfiable in $\Theta(D)$ dimensions but very close to the trivial baseline in $o(D)$ dimensions.

**Theorem 2.1.** *There exist constants $c_1, c_2$ such that, for every $D \geq 240$, $D = o(\sqrt{n})$ and every[3] $\varepsilon > 0$, with probability at least $\frac{1}{2} - o(1)$ the following happen simultaneously. For a random instance with $m = c_1 Dn$ constraints there exists an embedding in $\mathbb{R}^D$ that satisfies all the constraints. On the other hand, any embedding in $\mathbb{R}^d$, with $d = c_2 \varepsilon^2 D$, satisfies a fraction of at most $\frac{1}{2} + \varepsilon$ of the constraints.*

The theorem follows directly by Lemma 2.5 and Lemma 2.6, which prove the realizability of a random instance in $D$ dimensions and the accuracy collapse in $o(D)$ dimensions respectively, as described next.

**Realizability.** Notice that a priori, there is no guarantee that the random instance constructed as above is satisfiable, let alone embeddable in some low dimension with no errors. However, an interesting step in our analysis shows that if we were to use potentially *high* dimensions—much higher than $D$ and close to $n$—then we could satisfy all given triplets. To see this, based on the given triplets, we reduce the question of realizability in $n$ dimensions, to a question about the existence of *directed cycles* in random directed graphs sampled from an appropriate distribution: it suffices that the comparisons between distances do not induce any formal contradiction that corresponds to a *cycle* in an underlying directed graph whose vertex set consists of pairs $V \times V$. As we show in this regime, these random directed graphs contain no cycles, which in turn means that the random instance of geometric triplets (or quartets) is satisfiable in $n$ dimensions. Finally, to reduce the dimension needed from $n$ down to $D$, we show that the *arboricity*[4] of a suitable constraint graph is close to $D$, which then can be used to algorithmically find an embedding in $D$-dimensions satisfying all $m$ triplets (Avdiukhin et al., 2024).

**Accuracy Collapse.** For the second part in Theorem 1.3 we rely on a recent bound on the VC-dimension of contrastive learning proven in (Alon et al., 2024). In the framework of (Alon et al., 2024) a learning algorithm is given $m$ samples from a distribution $\mathcal{D}$ over $V^3 \times \{0, 1\}$: each sample is interpreted as a tuple $(x, y, z)$ along with a label, which indicates whether $\text{dist}(x, y) < \text{dist}(x, z)$ or $\text{dist}(x, y) > \text{dist}(x, z)$.

Consider the hypothesis class $\mathcal{H}$ of embeddings from $V$ to $\mathbb{R}^d$, where $d = \Theta(\varepsilon^2 D)$. Alon et al. (2024) prove that the VC-dimension of this hypothesis class is $\Theta(dn)$. Furthermore, observe that a random instance sampled from $\mathcal{I}(n, m)$ corresponds to a distribution $\mathcal{D}$ where the tuple is uniform among elements in $V^3$ and the labels are uniformly random. By the fundamental theorem of learning theory (Shalev-Shwartz & Ben-David, 2014), we have that with $m = \Theta(Dn) = \Theta(\frac{dn}{\varepsilon^2})$ samples, for every function in $\mathcal{H}$, the empirical risk is close to the true risk. That is, for every embedding $f : V \to \mathbb{R}^d$:

$$\left| \mathcal{R}(f) - \hat{\mathcal{R}}(f) \right| \leq \varepsilon, \tag{1}$$

where we use $\hat{\mathcal{R}}$ for the empirical risk and $\mathcal{R}$ for the true risk. On the one hand, the empirical risk corresponds to the fraction of constraints satisfied by the embedding, i.e., the triplet accuracy $\text{acc}(f)$. On the other hand, the true risk is the probability that an embedding is consistent with a random label, which is $1/2$, for every embedding. These two observations, together with the uniform convergence

---

[3] In fact, even for $D \leq c\sqrt{n}$ where $c$ is an absolute constant, our results hold with constant probability.

[4] Arboricity is a notion of graph density defined as $\rho(G) = \max_{H \subseteq V} \left\lceil \frac{|E(H)|}{|H| - 1} \right\rceil$.

bound of Equation (1) give us that no embedding can satisfy more than $\frac{1}{2} + \varepsilon$ of the constraints.

## 2.2. Realizability in $D$ dimensions

We first prove that a random instance of geometric triplets with $\Theta(Dn)$ constraints and $D = o(\sqrt{n})$ is satisfiable with high probability. We will consider a slightly different model for the random instance that is more convenient for our proofs and then reduce to $\mathcal{I}(n, m)$. In an instance sampled according to $\mathcal{I}(n, \lambda)$, for every $x, y, z \in V$ the number of occurrences of constraint $(x, y^+, z^-)$ follows a Poisson distribution with parameter $\lambda$, independent of everything else. Note that $m = \Theta(Dn)$ roughly corresponds to $\lambda = \Theta(\frac{D}{n^2})$ and the bound $D = o(\sqrt{n})$ corresponds to $\lambda = o\left(\frac{1}{n^{3/2}}\right)$.

We also describe a random model for directed (multi)-graphs again parameterized by $\lambda$, $\mathcal{G}_{\text{MAS}}(n, \lambda)$. We will show that a graph from this model is acyclic with high probability, which will imply that the triplets instance is realizable. This is a graph on $\binom{n}{2}$ vertices, i.e., the vertices of the graph are $\binom{[n]}{2}$ (each vertex will correspond to a distance between elements in $V$). For a pair of vertices $\{i, j\}$ and $\{k, l\}$ and for the directed edge $e = (\{i, j\}, \{k, l\})$, let $X_e$ be the random variable denoting the number of occurrences of edge $e$ in the multi-graph. Then $X_e = 0$ if $|\{i, j\} \cap \{k, l\}| = 0$ and $X_e \sim \text{Poi}(\lambda)$, independent of everything else, otherwise. We prove that if $\lambda$ is sufficiently small, a random graph generated from the aforementioned process is acyclic with high probability.

**Lemma 2.2.** *A graph* $G \sim \mathcal{G}_{MAS}(n, \lambda)$ *with* $\lambda = o(\frac{1}{n^{3/2}})$ *has no directed cycle with high probability.*

*Proof.* Let $Y$ be the random variable counting the number of directed cycles in $G$. Then we have that:

$$\mathbb{P}(G \text{ contains a directed cycle}) = \mathbb{P}(Y > 0)$$
$$\leq \mathbb{E}[Y],$$

where we have used the first moment method (Markov's inequality). We now let, for $k = 2, \ldots, n$, $Y_k$ be the random variable counting the number of directed cycles in $G$ with $k$ vertices, i.e., $Y = \sum_{k=2}^{n} Y_k$. We fix $k$ and bound $\mathbb{E}[Y_k]$. If $C_k$ is the set of directed cycles of size $k$, then by linearity of expectation:

$$\mathbb{E}[Y_k] = \sum_{C \in C_k} \mathbb{P}(C \in G).$$

Due to the generating process by which we are sampling the graph, $\mathbb{P}(C \in G)$ is not uniform over $C \in C_k$: for the majority of the cycles $C$ of size $k$ the probability that they appear in the graph is 0. In particular, consider a cycle $C = \{e_1, e_2, \ldots, e_k\}$ and let for notational convenience

$e_t = (\{i_t, j_t\}, \{k_t, l_t\})$. Note that if there exists a $t$ such that $|\{i_t, j_t\} \cap \{k_t, l_t\}| \neq 1$ then $\mathbb{P}(e_t \in G) = 0$ and thus $\mathbb{P}(C \in G) = 0$. Let $C_k^r$ be the set of cycles for which no such edge exists: these are the cycles that are realizable by our random generating process. On the one hand, the size of $C_k^r$ can be bounded by $\binom{n}{2}(2n)^{k-1} \leq (2n)^{k+1}$. On the other hand, for $C \in C_k^r$, we have that $\mathbb{P}(C \in G) = (1 - e^{-\lambda})^k$. We have that:

$$\mathbb{E}[Y_k] = \sum_{C \in C_k} \mathbb{P}(C \in G)$$
$$\leq (2n)^{k+1}(1 - e^{-\lambda})^k$$
$$= 2n(2(1 - e^{-\lambda})n)^k.$$

Using linearity of expectation and that $1 - e^{-\lambda} = \lambda + O(\lambda^2)$, we get that:

$$\mathbb{E}[Y] = \sum_{k=2}^{n} \mathbb{E}[Y_k]$$
$$\leq 2n \sum_{k=2}^{n} (2(\lambda + O(\lambda^2))n)^k$$
$$= 2n \frac{(2(\lambda + O(\lambda^2))n)^2}{1 - 2(\lambda + O(\lambda^2))n}$$
$$= o(1),$$

where we have used that $\lambda = o\left(\frac{1}{n^{3/2}}\right)$. $\qquad \square$

We can now prove that a triplet instance sampled from $\mathcal{I}(n, \lambda)$ is realizable.

**Lemma 2.3.** *An instance* $\mathcal{T} \sim \mathcal{I}(n, \lambda)$ *with* $\lambda = o\left(\frac{1}{n^{3/2}}\right)$ *is satisfiable in* $n$ *dimensions with high probability.*

*Proof.* We apply Lemma 3 from (Bilu & Linial, 2005). The instance is satisfiable in $n$ dimensions if and only if there exists a linear ordering of the distances consistent with the constraints. The question of satisfiability thus reduces to a question of the existence of cycles in a directed graph. We construct the directed graph $G$ as follows. The set of vertices of the graph corresponds to the set $\binom{V}{2}$ of size $\binom{n}{2}$. If $(x, y^+, z^-) \in \mathcal{T}$ then we add the directed edge $e = (\{x, y\}, \{x, z\})$. The linear ordering exists if and only if $G$ contains no directed cycles. Observe that $G \sim \mathcal{G}_{\text{MAS}}(n, \lambda)$ and by Lemma 2.2 the instance is satisfiable with high probability. $\qquad \square$

Next, we show how to embed the instance with no errors from $n$ down to $D$ dimensions. We prove the following upper bound on the arboricity $\rho(G)$ of random multi-graphs, which suffices for our purposes. We say that a multi-graph $G \sim \mathcal{G}(n, \lambda)$, if for every edge $e$ the number of occurrences of the edge follows a Poisson distribution with parameter $\lambda$, independent of everything else.

**Lemma 2.4.** *Let $G \sim \mathcal{G}(n, \lambda_n)$ with $\lambda_n = \frac{\alpha_n}{n}$ and $\alpha_n \geq 1$. Then, with probability $1 - o(1)$:*

$$\rho(G) \leq 5\alpha_n$$

We give the proof of the lemma in Appendix A.

We are now ready to prove the first claim of Theorem 2.1.

**Lemma 2.5.** *A random instance $\mathcal{T} \sim \mathcal{I}(n, m)$, with $m = c_1 Dn$ for $c_1$ being an absolute constant and $240 \leq D = o(\sqrt{n})$, is satisfiable in $D$ dimensions with high probability.*

*Proof.* We first prove the claim for an instance sampled from $\mathcal{I}(n, \lambda)$ and then reduce to an instance from $\mathcal{I}(n, m)$. Let $\mathcal{T} \sim \mathcal{I}(n, \lambda)$, with $D = o(\sqrt{n})$ and $\lambda = \frac{c_1' D}{n^2}$ (we will take $c_1' = 2c_1 = \frac{1}{120}$). Note that the conditions of Lemma 2.3 are satisfied and thus the instance is satisfiable in $n$ dimensions. We now consider the constraint (multi-)graph of the instance $G_c$, as defined in (Avdiukhin et al., 2024): we have $V(G_c) = V$ and for every $(x, y^+, z^-)$ we add the edges $\{x, y\}$ and $\{x, z\}$ to $G_c$. By Theorem 9 in (Avdiukhin et al., 2024), if the instance is satisfiable in $n$ dimensions then it is also satisfiable in $4\rho(G_c)$ dimensions. We now observe that we can write $G_c = G_1 \cup G_2$, where $G_1$ is the graph with edges coming from anchor-positive pairs and $G_2$ is the graph with edges from anchor-negative pairs. Observe that for $i = 1, 2$, $G_i \sim \mathcal{G}(n, \lambda')$ with $\lambda' = (1 + o(1))\frac{2c_1' D}{n}$. By Lemma 2.4 and the union bound we have that for $i = 1, 2$:

$$\rho(G_i) \leq 5(1 + o(1))2c_1' D \leq 15c_1' D.$$

We now have, by subadditivity of arboricity, that $\rho(G_c) \leq \rho(G_1) + \rho(G_2) \leq 30c_1' D$. By Theorem 9 in (Avdiukhin et al., 2024), the instance is satisfiable in $120c_1' D = D$ dimensions.

For the reduction to the $\mathcal{I}(n, m)$ model let $m = c_1 Dn$ with $c_1 = \frac{c_1'}{2}$. By the Chernoff bound (Exercise 2.3.5 in (Vershynin, 2018)) with probability $1 - o(1)$, for an instance $\mathcal{T} \sim \mathcal{I}(n, \lambda)$ it holds that $|\mathcal{T}| \geq m$. On the other hand, by Poisson conditioning (e.g. Theorem 3.7.8 in (Durrett, 2019)) conditioned on $|\mathcal{T}| = m'$, $\mathcal{T}$ has the same distribution as an instance sampled from $\mathcal{I}(n, m')$. Now consider an instance generated as follows. Sample $\mathcal{T}_1 \sim \mathcal{I}(n, \lambda)$ and if $|\mathcal{T}_1| \geq m$, then let $\mathcal{T}_2$ be an instance consisting of $m$ random constraints of $\mathcal{T}_1$, otherwise let $\mathcal{T}_2 = \mathcal{T}_1$. Note that since removing constraints cannot make a satisfiable instance unsatisfiable, we have that $\mathcal{T}_2$ is satisfiable in $D$ dimensions with high probability, and also $o(1)$-close in total variation distance to an instance sampled from $\mathcal{I}(n, m)$. Putting the two observations together yields the result. $\square$

## 2.3. Accuracy collapse in $d \approx \varepsilon^2 D$ dimensions

We now prove that with constant probability for a random instance of $m = \Theta(Dn)$ constraints, any embedding to $d \approx \varepsilon^2 D$ dimensions satisfies at most $\frac{1}{2} + \varepsilon$ of constraints. We show the following lemma:

**Lemma 2.6.** *Let $\mathcal{T} \sim \mathcal{I}(n, m)$ with $m = c_1 Dn$ and $c_1$ being an absolute constant. Then for any $\varepsilon > 0$, with probability at least $\frac{1}{2}$, for any embedding $f : V \to \mathbb{R}^d$, with $d = c_2 \varepsilon^2 D$ and $c_2$ being an absolute constant, it holds that it satisfies a fraction of at most $\frac{1}{2} + \varepsilon$ of constraints in $\mathcal{T}$.*

*Proof.* For this proof we will use the learning-theoretic framework of (Alon et al., 2024). We view embeddings $f : V \to \mathbb{R}^d$ as a hypothesis class $\mathcal{H}$ of functions $h : V^{\times 3} \to \{0, 1\}$. Intuitively, for an embedding $f$, for every $(x, y, z) \in V^{\times 3}$ either $(x, y^+, z^-)$ or $(x, z^+, y^-)$ is satisfied. In particular, for every embedding $f$ there is a corresponding hypothesis class $h_f$ such that for $(x, y, z) \in V^{\times 3}$:

$$h_f(x, y, z) = \begin{cases} 1, & \text{if } \|f(x) - f(y)\| < \|f(x) - f(z)\|, \\ 0, & \text{if } \|f(x) - f(y)\| > \|f(x) - f(z)\|. \end{cases}$$

By Theorem 3.2 in (Alon et al., 2024), there exists an absolute constant $c$ such that $\text{VC}(\mathcal{H}) \leq cnd$ where VC denotes the VC-dimension. We now consider distribution $\mathcal{D}$ over $V^3 \times \{0, 1\}$ that is the product of the distribution that is uniform over elements of $V^3$ where all three elements are distinct and the distribution that is uniform over $\{0, 1\}$. Note that an instance of $m$ samples from $\mathcal{D}$ has the same distribution as an instance sampled from $\mathcal{I}(n, m)$. We can now use Theorem 6.8 in (Shalev-Shwartz & Ben-David, 2014) to get that there exists a constant $C$ such that if $\mathcal{T} \sim \mathcal{I}(n, m)$ with $m \geq C\frac{\text{VC}(\mathcal{H})}{\varepsilon^2}$ then with probability at least $\frac{1}{2}$ for every embedding $f$ it holds that $\left| \text{acc}(f; \mathcal{T}) - \mathbb{P}_{(x,y)\sim\mathcal{D}} (h_f(x) = y) \right| \leq \varepsilon$. Now note that the label according to distribution $\mathcal{D}$ is random, meaning that for every $f$, $\mathbb{P}_{(x,y)\sim\mathcal{D}}(h_f(x) = y) = \frac{1}{2}$. Finally, note that $m = c_1 Dn = c_1 \frac{d}{c_2 \varepsilon^2}$, thus taking $c_2$ small enough so that $\frac{c_1}{c_2} \geq C$, we get that with probability at least $\frac{1}{2}$, for every $f$:

$$\left| \text{acc}(f; \mathcal{I}) - \frac{1}{2} \right| \leq \varepsilon,$$

which gives us the result. $\square$

**Extensions.** Our results can be extended to ordinal embeddings and the problem of quadruplet comparisons $(i, j, k, l)$. For our conclusions to hold, the upper bound required for the dimension is $D = o(n)$. In Appendix B, we describe the differences needed, and we also prove an analog of Lemma 2.2.

# 3. Inapproximability of Triplet Embeddings

Here we prove that approximating the accuracy in triplet embeddings better than the trivial baseline is hard:

**Theorem 3.1** (Approximate Triplet Embeddings). *Let $d \geq 1$. Assuming the Unique Games Conjecture, for every $\varepsilon > 0$, it is NP-hard to distinguish between triplet instances $(V, \mathcal{T})$ from the following two cases:*

1. *YES instance: There exists an embedding $f : V \to \mathbb{R}^d$ whose accuracy is $\mathrm{acc}(f; \mathcal{T}) \geq 1 - \varepsilon$.*

2. *NO instance: For every embedding $f : V \to \mathbb{R}^d$, $\mathrm{acc}(f; \mathcal{T}) \leq \frac{1}{2} + \varepsilon$.*

*Proof.* We provide an approximation-preserving reduction from Maximum Acyclic Subgraph (MAS) to the Triplet Embeddings problem. In MAS, we are given a directed graph $G(V, E)$ and we want to find a permutation of the vertices $\pi : V \to \{1, \ldots, |V|\}$, so as to maximize the number of directed edges $(u, v) \in E$ where $\pi(u) < \pi(v)$. Such edges are called satisfied, and we use $\mathrm{val}(\pi)$ to denote the fraction of satisfied edges by $\pi$ (where we normalize by $|E|$).

Observe that any MAS instance always admits a trivial solution such that at least $\frac{|E|}{2}$ directed edges are correctly oriented from left to right: simply output a random permutation on $V$. It is well-known by (Guruswami et al., 2008; 2011), that under Unique Games (Khot, 2002), instances of MAS are approximation resistant in the worst-case:

**Theorem 3.2** (Guruswami et al. (2008)). *Assuming Unique Games, for every $\varepsilon > 0$, it is NP-hard to distinguish between MAS instances from the following two cases:*

1. *YES instance: There exists a permutation $\pi^*$ whose value $\mathrm{val}(\pi^*) \geq 1 - \varepsilon$.*

2. *NO instance: For every permutation $\pi$, $\mathrm{val}(\pi) \leq \frac{1}{2} + \varepsilon$.*

Below we describe the gap reduction from MAS to triplet embeddings, and show how to map YES instances of MAS to YES instances of triplet embeddings, and similarly, NO instances of MAS to NO instances of triplet embeddings.

**Gap Reduction.** Let $G = (V, E)$ be a directed graph which is the input to MAS. We construct a triplet instance $(U, \mathcal{T})$ as follows:

- **Items.** We create $|V| + 1$ items by introducing a new distinguished anchor item $S$ and let $U := V \cup \{S\}$.

- **Triplet Comparisons.** We create $|\mathcal{T}| := |E|$ triplets: for every directed edge $(u \to v) \in E$, we introduce the triplet $(S, u, v) \in \mathcal{T}$, i.e., all triplets are with respect to the same anchor item $S$.

Moreover, given any embedding $f : U \to \mathbb{R}^d$, we define the *radius* of each vertex $v \in V$ to be its distance from the anchor:

$$r_f(v) := \|f(v) - f(S)\|_2$$

Let $\pi_f$ be any total order of $V$ obtained by sorting vertices by increasing $r_f(v)$ (breaking ties arbitrarily). Then for every edge $(u \to v) \in E$, the triplet $(S, u, v)$ is satisfied by $f$ if and only if $r_f(u) < r_f(v)$ which is equivalent to $\pi_f(u) < \pi_f(v)$.

Therefore,

$$\mathrm{acc}(f; \mathcal{T}) = \frac{|\{(u \to v) \in E : \pi_f(u) < \pi_f(v)\}|}{|E|}.$$

Conversely, for any total order $\pi$ of $V$, define a one-dimensional embedding $f_\pi : U \to \mathbb{R}$ by

$$f_\pi(S) = 0, \qquad f_\pi(v) = \pi(v) \quad \text{for all } v \in V.$$

Then for every edge $(u \to v) \in E$, the triplet $(S, u, v)$ is satisfied by $f_\pi$ if and only if $\pi(u) < \pi(v)$. Hence,

$$\max_{f:U \to \mathbb{R}^d} \mathrm{acc}(f; \mathcal{T}) = \max_\pi \frac{|\{(u \to v) \in E : \pi(u) < \pi(v)\}|}{|E|}$$

and the optimum is attained already in one dimension. $\square$

**Extensions.** We note that the same hardness of approximation holds for ordinal embeddings more generally, i.e., for the case of quadruplet comparisons of the form $(i, j, k, l)$ indicating that $\|f(i) - f(j)\|_2 < \|f(k) - f(l)\|_2$. This follows as triplet instances are special cases (replace every $(i, j, k)$ triplet with the quadruplet $(i, j, i, k)$).

# 4. Experiments

We provide both synthetic and real-world data experiments to support Theorem 1.3, which predicts that when the embedding dimension $d$ falls below a constant fraction of the ground-truth dimension $D$, accuracy drops significantly and becomes comparable to the trivial $50\%$-baseline. Notice that in the context of text embeddings and retrieval, a similar drop was observed under aggressive dimension truncation in (Takeshita et al., 2025; Tsukagoshi & Sasano, 2025). Here, we do not truncate the dimension, but rather optimize to find the best embedding under a constraint on its dimension $d$ (which we call the target dimension).

**Data.** We sample $n = 1000$ points independently and uniformly from the unit sphere in $\mathbb{R}^D$, where $D \in \{128, 256, 512, 1024\}$. Triplets $(i, j, k)$ are sampled uniformly without replacement and labeled by the ground-truth Euclidean distance. Here, we fix the number of triplets to $m = 10^6$. Observe, that even though this is a dense instance, by construction, these instances are realizable in

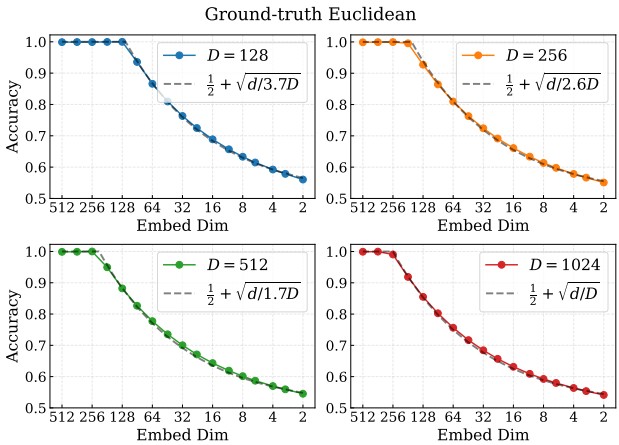

*Figure 1.* Ground-truth Euclidean embeddings: empirical accuracy vs. theoretical prediction $\frac{1}{2} + \sqrt{d/(c_2 D)}$ (clipped at 1) with $n = 1000$, $m = 10^6$.

dimension $D$ with perfect accuracy. As we vary $D$, and the target dimension $d$, we measure the triplet accuracy, i.e., the fraction of satisfied triplets by the found embedding (see also training procedure below).

**Embeddings and Training Procedure.** For a fixed target embedding dimension $d$, we directly optimize embeddings $f : V \to \mathbb{R}^d$ using AdamW (Loshchilov & Hutter, 2019) with the standard hinge triplet loss (Schroff et al., 2015):

$$\mathcal{L} = \max\left(0, \|f(i) - f(j)\|_2^2 - \|f(i) - f(k)\|_2^2 + \gamma\right)$$

where the margin parameter is set to $\gamma = 1$.

For each ground-truth dimension $D$ we consider from $\{128, 256, 512, 1024\}$, we sweep the embedding dimension $d$ over powers of two in $[2, 512]$ and include intermediate geometric means between consecutive powers to obtain finer resolution.

**Plots.** In Figure 1, the y-axis is the triplet accuracy defined in Def. 1.1, and the x-axis has various target dimensions $d$. The accuracy decreases towards the random baseline for sufficiently small target dimension. Moreover, the empirical accuracy curve[5] closely tracks the theoretical prediction, i.e., accuracy $\approx \frac{1}{2} + \sqrt{d/(c_2 D)}$ (dashed line in each panel) for a per-$D$ fitted constant $c_2$. Notably, $c_2$ does not need to be extremely small for the predicted decrease to kick in, suggesting that Lemma 2.6 may hold over a wider range of parameters than its proof requires.

**Real-world data.** We evaluate two real-world embedding sources at $D = 384$, $n = 1000$, $m = 10^6$: text embeddings

---

[5]All results are averaged over 5 random seeds; standard deviations are below 0.005 in all settings and are therefore not visible at the scale of the figure.

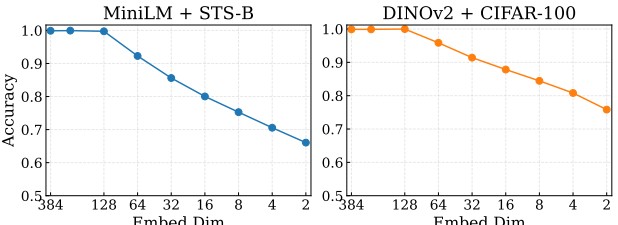

*Figure 2.* Real-world triplet accuracy on STS-B and CIFAR-100.

from MiniLM (Wang et al., 2020) on STS-B, and image embeddings from DINOv2 (Oquab et al., 2023) on CIFAR-100. The triplets are sampled uniformly and labeled by the Euclidean distance in the source 384-d embedding space. As expected and as we observed experimentally (Figure 2), real-world data contains structure that potentially allows for better accuracy at lower dimensions. We believe that the less pronounced collapse here reflects exploitable structure in real embeddings, and is an interesting future research direction on beyond-worst-case contrastive embeddings: essentially, the random model serves as a calibration baseline, and the gap between random and real-data accuracy quantifies the degree of exploitable structure of a dataset (e.g., triplets generated in non-adversarial/non-random way, large-margin assumptions, latent clustered structure etc.).

## Conclusion

We investigated the fundamental role of dimensionality in embedding-based representations. Our main results are both information-theoretic and computational. We first examined standard contrastive embedding tasks based on triplet comparisons and proved that dimensionality acts as a bottleneck for accuracy: even when triplet constraints are perfectly realizable in high dimension, reducing the embedding dimension below a constant fraction of the ground-truth dimension can force accuracy to collapse down to the trivial $\frac{1}{2}$-baseline. We further demonstrated that this limitation persists at the algorithmic level, establishing strong hardness of approximation results under standard complexity assumptions. It would be very interesting to understand how imposing extra structure on the input, e.g., separability conditions, large-margin assumptions, may allow us to bypass negative results, and obtain significantly better-than-random accuracy, even in low dimensions (independent of $n$).

## Acknowledgments

We would like to thank the anonymous ICML reviewers for their useful feedback and suggestions. We would like to thank Konstantin Makarychev for valuable discussions throughout this project. D. Arvanitakis was supported by NSF Awards CCF-1955351 and EECS-2216970, and in part

by grants from the NSF (DMS-2235451) and the Simons Foundation (MPS-NITMB-00005320) to the NSF-Simons National Institute for Theory and Mathematics in Biology (NITMB). V. Chatziafratis and Y. Luo were supported by a UC Santa Cruz start-up grant and by Hellman's fellowship.

## Impact Statement

This paper presents work whose goal is to advance the field of Machine Learning. There are many potential societal consequences of our work, none which we feel must be specifically highlighted here.

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

# Appendix

## A. Omitted Proofs

*Proof of Lemma 2.4.* We show that $\max_H \frac{|E(H)|}{|H|-1} \leq 4\alpha_n$, applying the ceiling function, we have that $\rho(G) = \left\lceil \max_H \frac{|E(H)|}{|H|-1} \right\rceil \leq \lceil 4\alpha_n \rceil \leq 4\alpha_n + 1 \leq 5\alpha_n$. We have, by the union bound that:

$$\mathbb{P}\left(\exists H : \frac{|E(H)|}{|H|-1} \geq 4\alpha_n\right)$$

$$\leq \sum_{k=2}^{n} \mathbb{P}\left(\exists H : |H| = k \text{ and } E(H) \geq 4(k-1)\alpha_n\right).$$

Fix $k$ and $H \subseteq V$ with $|H| = k$ and observe that the number of edges in $H$ follows a Poisson random variable with parameter $\binom{k}{2} \cdot \lambda$, applying the Chernoff bound for Poisson tails (Exercise 2.3.3 in (Vershynin, 2018)) we have that:

$$\mathbb{P}\left(|E(H)| \geq 4(k-1)\alpha_n\right) \leq \left(\frac{ek}{8n}\right)^{2k},$$

where we have used that $\alpha_n \geq 1$ and $k \geq 2$. Taking the union bound over all subset of $V$ of size $k$, we have that:

$$\mathbb{P}\left(\exists H : |H| = k \text{ and } E(H) \geq 4(k-1)\alpha_n\right)$$
$$\leq \left(\frac{en}{k}\right)^k \left(\frac{ek}{8n}\right)^{2k}$$
$$\leq \left(\frac{e^2 k}{8n}\right)^k.$$

It remains to show that $\sum_{k=2}^{n} \left(\frac{e^2 k}{8n}\right)^k = o(1)$. We have that:

$$\sum_{k=2}^{n} \left(\frac{e^2 k}{8n}\right)^k = \sum_{k=2}^{\sqrt{n}} \left(\frac{e^2 k}{8n}\right)^k + \sum_{k=\sqrt{n}+1}^{n} \left(\frac{e^2 k}{8n}\right)^k$$
$$\leq \sum_{k=2}^{\sqrt{n}} \frac{1}{n^2} + n \left(\frac{e^2}{8}\right)^{\sqrt{n}}$$
$$= o(1)$$

$\square$

## B. Extension to quadruplet comparisons

In this section we describe how our results can be extended to the problem of quadruplet comparisons (also called geometric quartets). We again assume that there is a set of elements $V$ with $|V| = n$ and we are given a set of $m$ geometric quartet constraints of the form $(x, y, z, w)$. The goal is to embed the elements of $V$ through a map $f$ to $\mathbb{R}^d$ so as to satisfy as many quartet constraints as possible. We say that $f$ satisfies a constraint $(x, y, z, w)$ if $\|f(x) - f(y)\| < \|f(z) - f(w)\|$, we say that an instance is satisfiable in $d$ dimensions if there is an $f : V \to \mathbb{R}^d$ that satisfies all the constraints. We again consider random instances sampled from a distribution $\mathcal{I}_Q(n, m)$ consisting of $m$ constraints, $(x_i, y_i, z_i, w_i)$ selected uniformly at random in $V^4$.

**Realizability** We consider an instance with $m = Dn$ constraints and $D = o(n)$. We begin by observing that the algorithm of (Avdiukhin et al., 2024) can be extended to work for the quadruplet reconstruction problem. The only difference is in the construction of the constraint graph $G_c$. In this setting, for a constraint of the form $(x, y, z, w)$ we add the edges $\{x, y\}$ and

$\{z, w\}$, we can then proceed with the algorithm of (Avdiukhin et al., 2024), which gives us that the instance can be satisfied in $4\rho(G_c)$ dimensions given that the instance is satisfiable in $n$ dimensions.

We turn our attention to an instance $\mathcal{I}_Q(n, \lambda)$, $\lambda = \frac{D}{n^3} = o(\frac{1}{n^2})$ where for every $x, y, z, w \in V$ the occurrences of constraint $(x, y, z, w)$ follows a Poisson with parameter $\lambda$. In that case, the constraint graph follows the distribution $\mathcal{G}(n, 2\lambda')$ with $\lambda' \approx 4\frac{D}{n}$, which gives us, by Lemma 2.4 that the arboricity is at most $\Theta(\lambda'n) = \Theta(D)$. We can thus conclude that if the instance is satisfiable in $n$ dimensions then it is also satisfiable in $\Theta(D)$ dimensions. It therefore, remains to show that the instance is satisfiable in $n$ dimensions. By Lemma 3 in (Bilu & Linial, 2005) this is reduced to showing that a random directed graph $G$ where the set of vertices is $\binom{[n]}{2}$ and every directed edge $(\{i, j\}, \{k, l\})$ occurs $\mathrm{Poi}(\lambda)$ number of times contains no directed cycle. This is proven in the following lemma.

**Lemma B.1.** *Let $G$ be a random graph with vertex set $\binom{[n]}{2}$ and where for every $e = (\{i, j\}, \{k, l\})$ the directed edge occurs $X_e \sim \mathrm{Poi}(\lambda)$, with $\lambda = o(\frac{1}{n^2})$*

*Proof.* Let $X$ be the number of directed cycles in $G$, we have that:

$$\mathbb{P}\left(G \text{ contains a directed cycle}\right) = \mathbb{P}\left(X > 0\right)$$
$$\leq \mathbb{E}\left[X\right],$$

where we have used the first moment method (Markov's inequality). We now let $X_k$ to denote the number of directed cycles of size $k$, $X = \sum_{k=2}^{n} X_k$, for the expectation of $X_k$ we have, by linearity of expectation:

$$\mathbb{E}\left[X_k\right] = \binom{\binom{n}{2}}{k}(k-1)!\left(1 - e^{-\lambda}\right)^k,$$

where $\binom{\binom{n}{2}}{k}(k-1)!$ is the total number of directed cycles and $\left(1 - e^{-\lambda}\right)^k$ is the probability that a particular directed cycle is realized in the graph. This in turn, using that $1 - e^{-\lambda} = \lambda + O(\lambda^2)$, gives us that:

$$\mathbb{E}\left[X\right] \leq \sum_{k=2}^{\binom{n}{2}} \binom{\binom{n}{2}}{k}(k-1)!(\lambda + O(\lambda^2))^k$$
$$\leq \sum_{k=2}^{\binom{n}{2}} \left((\lambda + O(\lambda^2))n^2\right)^k$$
$$\leq ((\lambda + O(\lambda^2))n^2)^2 \frac{1}{1 - (\lambda + O(\lambda^2))n^2}$$
$$= o(1),$$

where we have used that $\binom{\binom{n}{2}}{k}(k-1)! \leq n^{2k}$

**Accuracy collapse**   To show accuracy collapse for quadruplet reconstruction, we observe that the proof of (Alon et al., 2024) applies verbatim and yields a VC-dimension of $\Theta(dn)$, where $d$ is the embedding dimension. We then follow the proof of Lemma 2.6. $\square$

## C. Additional Experiments

### C.1. Normalized Embeddings

We repeat the synthetic experiment of Section 4 with the modification that, after each optimization step, all embeddings are normalized to unit vectors. This resembles cosine-similarity-based representation learning. Results are shown in Figure 3.

### C.2. Uniformly Random Triplets

We also generate instances with no explicit ground-truth by sampling $m = 10^6$ triplets uniformly at random over $n = 4000$ items, assigning each comparison independently. For sufficiently large $d$, gradient-based optimization achieves $\mathrm{acc} = 1$, so these instances are realizable. This gives a second source of realizable triplet constraints without specifying a ground-truth geometry. Results are shown in Figure 4.

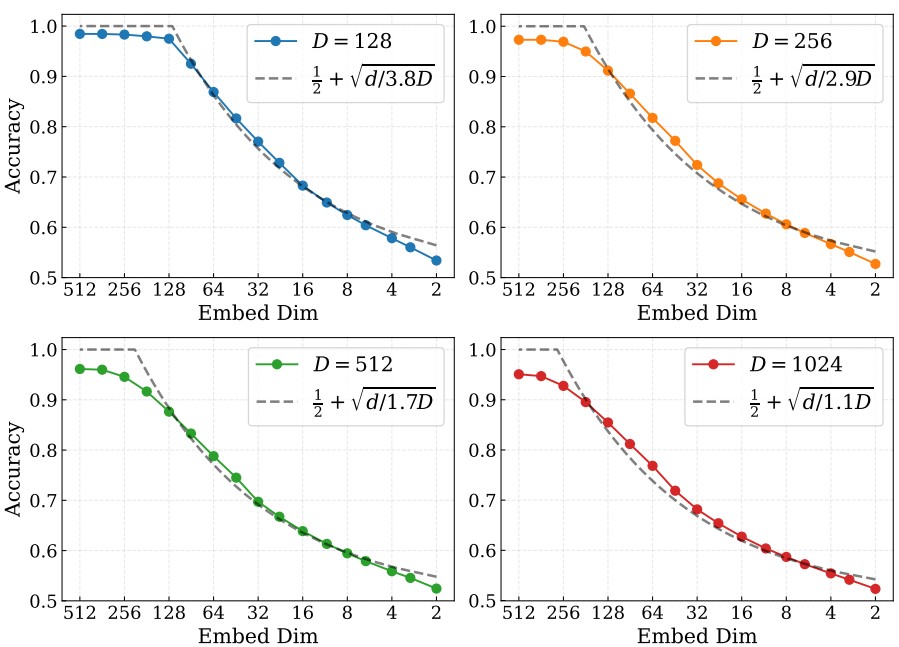

*Figure 3.* Normalized embeddings on synthetic ground-truth Euclidean data. Same setting as Figure 1 but with all embeddings normalized to unit vectors after each gradient step.

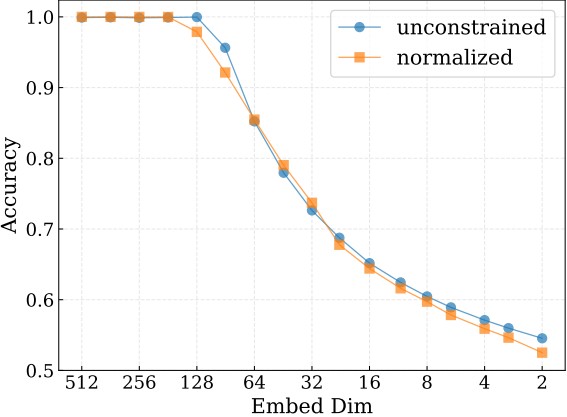

*Figure 4.* Uniformly random triplets with no explicit ground-truth: triplet accuracy under various target embedding dimensions $d$.

