# OpenReview forum: "Provable Accuracy Collapse of Embedding-Based Representations under Dimensionality Mismatch"
_ICML.cc/2026/Conference — ICML 2026 spotlight_

### Official Review · Reviewer_PMwt · 2026-03-11

**Soundness:** 3
**Presentation:** 3
**Significance:** 3
**Originality:** 2
**Overall Recommendation:** 5
**Confidence:** 3

**Summary:**

This paper studies metric embeddings that should satisfy as many as possible of a given set of triplet comparisons $(i,j,k)$, which means that $i$ should be embedded closer to $j$ than to $k$.

The problem is well motivated in the machine learning context.

There are two main results:
1. *An impossibility result in the realizable case*: for any $D$ and $\varepsilon>0$ there are worst-case instances such that all triplets can be satisfied in $\mathbb{R}^D$; however, in dimension $c_\varepsilon D< D$ only $1/2+\varepsilon$ triplets can be satisfied (where $1/2$ can be achieved by a simple random embedding).
2. *A computational hardness result for the agnostic case*: Under standard complexity-theoretic assumptions (unique games conjectures) it is hard to approximately maximize the number of satisfied triplets.

Additionally, there are some experiments.

**Compliance With Llm Reviewing Policy:**

Affirmed.

**Final Justification:**

Having read the authors responses and the authors' rebuttal I believe this paper is strong and should be accepted.

**Key Questions For Authors:**

Are there interesting and/or known special cases, where the triplets are realizable by some special family of metrics (not necessarily Euclidean, e.g., tree metric, etc.) or other such assumptions allowing such embeddings? Basically I am asking for positive results, where the lower bounds obtained here do not hold.

How tight/optimal is Thm 1.3? In particular, is $c_\varepsilon = O(\varepsilon^2)$ necessary?

Can you make your result "dense"? I.e., can you design instances such that $\mathcal{T}=V^3$ gives the lower bound? I believe this would make the results more meaningful (and less worst-case).

What about other metrics (tree metrics, $\ell_p$ metrics, ...)? This might allow a stronger practical motivation?

About your future work on comment on incorporating margin assumption: The problem seems to become easily solvable with, e.g., the JL lemma and $\varepsilon=\gamma/2$ assuming a margin of $\gamma$. Please comment.

**Limitations:**

yes

**Strengths And Weaknesses:**

This paper is nicely written and easy to follow. It tackles a well motivated and important problem. Related work is discussed, making it clear that standard approaches (e.g. JL lemma) are not sufficient here.

The two main theoretical results (impossibility and computational hardness) are interesting and should be interesting for a subcommunity at ICML. The technical contributions are non-trivial. However, at times it seems like most of the results follow directly from Avdiukhin et al., [2024] and Alon et al. [2024]. *This is probably my main critique*: contrast your own technical contribution more clearly with these two particular papers.

The exposition could be improved further:
* You could add some convincing known or own examples for special cases (of metrics/additional structure) that makes this problem solvable. While you point to some references it is not clear how general or interesting these examples are form the perspective of learning.

Please, also see the questions below.

---

> ### Author Rebuttal · Authors · 2026-03-31
>
> We thank the reviewer for the positive score and useful feedback. We answer their questions:
> ___
> *Question:  ...interesting and/or known special cases with positive results?*
>
> Answer: Obtaining positive results in the context of provable contrastive learning has been difficult. To the best of our knowledge, one of the current best results is to obtain a PTAS, under density assumptions, and only for the case of dimension  $D=1$, i.e., for line embeddings. In that work the authors assume they have access to $\Omega(n^3)$ triplets and they want to embed points on the line. More generally, the ideal result here would be to algorithmically obtain a low-dimensional ordinal embedding that satisfies many of the given triplets. Of course, this can happen in special cases, for example, if the given triplets are produced by a line embedding, but in general the tradeoffs between accuracy and dimension are not well-understood. Studying such questions under the beyond-worst-case lens is very interesting.
>
> [1] Fan et al. "Learning Lines with Ordinal Constraints", APPROX 2020
>
> Regarding other types of metrics, we are aware of a positive result in the context of ultrametric embeddings [2]. There they show how with $n\ log n$ adaptive triplet queries, one can reconstruct the underlying hierarchical clustering. We note that the question for tree metrics needs to be adapted, as there is no notion of dimensionality of a tree. However, perhaps the depth of the tree can be used as the parameter of interest, when evaluating the triplet comparisons.
>
> [2] Emamjomeh-Zadeh, Ehsan, and David Kempe. "Adaptive hierarchical clustering using ordinal queries."  SODA 2018.
> ___
> *Question: How tight/optimal is Thm 1.3?*
>
> Answer: Please see our response above to Reviewer iVZF who had the same question: This is an interesting question. The ideal/strongest result here, that would highlight the importance of choosing the right $d$ would be if we could have a construction of a dataset, that is realizable with $D$ dimensions (same as part 1 of our Theorem 1.3), yet *every* embedding of dimension $d=D-1$ already fails to achieve accuracy better than the random baseline of $1/2$. However, we are not aware of such constructions that allow for such sharp tradeoffs, and in fact, we believe that such strong statements do not hold for the task of contrastive learning from triplets, studied in our work. We believe it is an interesting direction to find constructions where $c(\epsilon)$ is very close to 1.
>
> In our result, the stated bound comes from using the fundamental learning theory result for uniform convergence, and for our approach this bound is tight, i.e., having $c(ε)\approx \epsilon^2$ is the best that we can obtain. There could potentially be some other more clever construction, that yields better dependence on $d$ vs $D$. We note that experimentally, we verified this dependence for our construction, as the predicted accuracy $Acc = 1/2 + \sqrt{d/D}$ nearly matches the observed behavior in several contexts.
>
> Another interesting question along these lines, is to understand the benefits of dimensionality in contrastive learning in low-dimensions, for example by proving separations in low-dimensions, i.e., constructing datasets which are realizable for some small dimension say $d$, yet any embedding in lower dimensions say $d'< d$ achieves very bad accuracy.
> ___
> *Question: On ``dense'' instances...?
>
> Answer: This is a great question! One approach perhaps to extending our result to dense instances, could be to start from a random instance realizable in $D$ dimensions, as we do in our main result; and then including all remaining triplets yielding a final input of $\Theta(n^3)$ triplets. This ensures the realizability (given that the original triplets are sufficiently sparse). However, after padding the input with the remaining triplets, it is not clear how to show that every other embedding in lower dimensions $d= c D$ only gets $1/2$ or very close to $1/2$. We actually believe that such a direct construction may not be possible.
>
> ___
> *Question: Other metrics (tree metrics, $\ell_p$ metrics, ...)?*
>
> Answer: We believe that it would be possible to extend some of our constructions/analyses for other $\ell_p$, particularly for $\ell_1$ and $\ell_\infty$ with a potentially worse dependence on the approximation between $d$ and $D$. Regarding tree metrics, as mentioned, there is not a direct notion of dimensionality of the tree, please see our response above to your 1st question.
> ___
> *Question: On incorporating margin assumption...*
>
> Answer: In the section where we discussed future directions, we were interested in avoiding the dependence on $n$ under some structural assumptions. You are right that under the large-margin assumption, if we are fine with a logarithmic dependence, then JL could work.

---

> > ### Author Rebuttal · Reviewer_PMwt · 2026-04-02
> >
> > Thank you for these comments. I raised my score.

---

### Official Review · Reviewer_CDfM · 2026-03-12

**Soundness:** 3
**Presentation:** 2
**Significance:** 3
**Originality:** 3
**Overall Recommendation:** 5
**Confidence:** 4

**Summary:**

This paper investigates theoretically how well triplet constraints, like in contrastive learning, can be preserved as the dimensionality of the embedding space decreases. There are two main results. First, for many sets of triplet constraints the number of satisfiable constraints drops to only half at a constant multiple of the dimension in which all constraints can still be satisfied. Second, deciding whether this collapse happens for a specific set of constraints is NP-hard under the Unique Games Conjecture. The authors run some toy experiments to illustrate their findings.

**Compliance With Llm Reviewing Policy:**

Affirmed.

**Final Justification:**

This is a good theory paper on a topic relevant to the broad field of contrastive learning. The rebuttal improved the paper a lot with several additional experiments, which in part also highlight the limitation of the considered random triplet setting. My concerns are properly addressed and I increased my score to 5.

**Key Questions For Authors:**

Q1: How realistic is the random model of constraints that the authors use in chapter 2? I would expect that constraint sets that actually come up in machine learning have more structure, e.g., come from a geometric configuration in $\mathbb{R}^D$ and the anchor and positive sample come from points that a very nearby, while negative samples are very far from the anchors. Triplets where positive and negative samples are about equally far from the anchor are probably more rare. This may be relevant as Thm 2.1 only holds with probability 0.5 for the specific random model employed. So many real world constraint sets might fall in the case where the Theorem does not apply.

Q2: At $d=2$ most of the simulations are essentially at chance level. Nevertheless, contrastive visualization methods like TriMap, PacMAP, or UMAP explicitly optimize an embedding given triplet constraints and often achieve informative visualization of data whose intrinsic dimensionality is much larger (thus likely $D\gg 2$). How does this fit together? If there is catastrophic accuracy collapse and triplet constraints are at chance level, how can the visualization be informative? And vice versa: If the visualizations are informative, maybe triplet constraint accuracy of about 0.5 might not be so problematic.

Q3: Avdiukhin et al. also considered kNN embeddability, a special case of triplet constraints. However, I guess to encode a kNN graph more than $m=\Theta(Dn)$ constraints are needed, so that the current theory from Section 2 does not apply. Do the authors have an idea of how to extend their results to kNN graphs? This would be quite relevant as kNN recall is a common metric for visualization methods.

**Limitations:**

Limitations are not explicitly discussed. Especially real-world applicability should be addressed.

**Strengths And Weaknesses:**

**Soundness**
I did not check the proofs in the appendix.

*Pro*
Overall, the key part of the paper, the theoretical statements, seem valid, though I have a few questions, see below. It is nice that there is at least some empirical illustration.


*Con*
The theorem on collapse (Thm 2.1) holds only for constraints that are embeddable in $D\geq 240$ dimensions. This is very high and arguably many real world datasets have at least intrinsic dimensionality below $240$.

There are no uncertainties in the experiments. Such toy experiments should be easy to repeat (or in case these experiments were repeated, how many runs were conducted?). In addition, since Thm 2.1 only holds with probability about 0.5, it would be interesting to see the distribution of accuracy per embedding dimension. At least half the cases should follow Thm 2.1.

It would strengthen the paper a lot if some experiments also used real data. E.g. some real constraint set of a contrastive algorithm, or the constraints encoded in some embedding created with a contrastive learning algorithm. One could work out both $D$ (given in the second case) and compute the accuracy for various $d$. This would help to evaluate how problematic the the collapse in accuracy is on real data.


Line 227 right: Should the middle inequality not be a $\leq$?

Line 230 left: $d=o(\sqrt{n})$ should be $D=o(\sqrt{n})$, I think.


I have some questions on the theory:

- Why is the probability in Thm 2.1 only $0.5-o(1)$? If I understand the proof correctly, the probability of $0.5$ only comes from Lem 2.6 and there is no $-o(1)$.

- I do not quite understand the multi-graph setup. E.g. if an edge (i.e. a constraint) is present multiple times, does it count multiple times to the accuracy competition? When counting directed cycles, are there multiple different cycles that only differ in which parallel edge was chosen?

- Why chose the lose bound of $10(1+o(1)) \leq 15$? Would a tighter bound not improve on the minimal dimensionality $D$ at which Thm 2.1 holds.

- It is unclear to me if satisfiability means satisfiable in the Euclidean distance or any distance on $\mathbb{R}^D$?




**Presentation**
*Pro*
Although most of the proofs are part of the main paper, the authors also give high-level descriptions of their proof strategies, which helps with accessibility.

*Con*
The arguments for Theorem 2.1 were quite unintuitive for me. Perhaps one can convey some illustration graphically?

It would be helpful to add a curve showing the predicted $0.5 + \varepsilon$ for various $(D, d)$ combinations in Figures 1 and 2 to showcase how well the theory predicts the experimental output. To avoid clutter, this might mean more panels, but the current ones are also very large for the content they show.

Clarify that the probability 1/2 primarily applies to the second part of the statement of Thm 2.1 (poor embedding in $\mathbb{R}^d$). Currently, it seems to primarily apply to the first part (statisfiability in $\mathbb{R}^D$), which if I understand correctly, holds with high probability.

The introduction is nearly 3 pages long, which seems excessive.


**Significance**
*Pro*
Contrastive learning with triplet constraints is very prominent (although constraints are often implicitly generated on the fly). This makes the theoretical contribution of the paper important. It is more practically relevant than prior works on minimally needed dimensionalities for full statisfiability. The presented theory also seems to be able to explain some recent empirical findings, underlining its timeliness.

*Con*
A minimal embedding dimension of $D=240$ seems very high. In some cases it is unclear how much the theory applies to real world contrastive learning setups, where constraint sets are not generated through a random process.


**Originality**
*Pro*
The results build in part on earlier work by Avdiukhin et al. and Alon et al., but extend their ideas and apply them to a novel questions. I think this paper is quite original.

---

> ### Author Rebuttal · Authors · 2026-03-31
>
> We thank the reviewer for their constructive feedback. Our responses:
> ___
> *Question: ...$D\ge 240$?*
>
> Answer:  The stated constant can be improved. In Lemma 2.5, we chose certain constants nicely, because we wanted to focus on clarity rather than optimizing that constant. Specifically, 240 arises from bounding arboricity (factor 120 in the proof) and ensuring the random graph is acyclic. The true threshold is a bit lower, and our experiments (collapse for $D$ as small as 128) support this. We also note, that real-world embedding models operate above this dimension (OpenAI has $D = 3072$, Gemini $D = 768$ to $3072$.)
> ___
> *Question: Add a curve showing the predicted $0.5 + ε$ for various $(D, d)$...*
>
> Answer:
> Thank you for this important suggestion. To clarify the setting, we have repeated all experiments in the paper with five random seeds $\{0,1,2,3,4\}$. We also tuned the training hyperparameters, which yields slightly higher accuracy over the original submission. Now, we present the accuracy results for different choices of $d$ (embedding dimension) for a dataset where the ground-truth dimension is $D=1024$. We also present the theoretical prediction which is $\frac{1}{2}+\sqrt{d/c_2 D}$. Surprisingly, with the constant $c_2 = 1$, the theoretical bound nearly matches the observed results , thus informing us on how the accuracy changes when increasing $d$.
>
> | Dimension d | Empirical Mean $\pm$ Std | Theory Prediction $\frac{1}{2}+\sqrt{d/c_2 D}$ |
> | :--- | :---: | ---: |
> | 4 | $.5637 \pm .0003$ | $.5625$ |
> | 8 | $.5925 \pm .0008$ | $.5884$ |
> | 16 | $.6314 \pm .0004$ | $.6250$ |
> | 32 | $.6843 \pm .0005$ |  $.6768$ |
> | 64 | $.7563 \pm .0006$| $.7500$ |
> |128 | $.8551 \pm .0007$ | $.8536$ |
> | 256 | $.9906 \pm .0009$ |  $1.0$ |
> | 512 | $.9993 \pm .0004$|  $1.0$ |
>
> Takeaway: The rightmost column (theory prediction) nearly matches the middle column (empirical observations).
> ___
> *Question: Regarding real-world data and exploiting the underlying structure...*
>
> Answer: We agree that real-world constraints have more structure than our random model. For structured constraints, additional structural assumptions could potentially circumvent the collapse. To quantify this, we performed two new experiments with real-world data, one for text embeddings and one for images with $D{=}384$, $n{=}1000$ and sampled $m{=}10^6$ triplets. Results shown in below (averaged over random seeds $\{0,1,2,3,4\}$, let us know of any questions).
>
> | d | MiniLM Mean + STS-B (Mean $\pm$ Std) | DINOv2+CIFAR100 (Mean ± Std)|
> | :--- | :---: | ---: |
> | 4 | $.7056 \pm .0029$ | $.8080 \pm .0018$ |
> | 8 | $.7527 \pm .0034$| $.8445 \pm .0009$ |
> | 16 | $.8003 \pm .0035$| $.8784 \pm .0009$ |
> | 32 | $.8560 \pm .0032$ |   $.9141 \pm .0007$ |
> | 64 | $.9227 \pm .0019$ |$.9586 \pm .0005$ |
> |128 | $.9974 \pm .0010$ | $.9989 \pm .0001$ |
> | 256 |$.9994 \pm .0003$ |  $.9999 \pm .0005$ |
> | 384 (ground truth $D$) | $.9999 \pm .0005$|  $.9999 \pm .0004$ |
>
> Takeaway: Our theoretical bound on the random model serves as a *calibration baseline*: among all triplet instances realizable in $D$ dimensions, random instances seem to have little exploitable structure. Indeed, the observed accuracy gap quantifies how much low-dimensional structure a given dataset possesses. For real-data, accuracy exceeds the worst-case bound, confirming that structure present in real embeddings can mitigate the accuracy collapse predicted by Theorem 2.1. Finally, the fact that DINOv2 consistently outperforms MiniLM at every $d$ aligns with the well-known lower intrinsic dimensionality of natural images compared to text.
> ___
> *Question: ...Is it Euclidean distance or any distance on $R^D$?*
>
> Answer: Yes, all results use Euclidean distance (Def. 1.1).
> ___
> *Question: On k-NNs, and methods like TriMap, PacMAP, or UMAP...*
>
> Answer: For the k-NN setting, the key distinction is in *which triplets* are being evaluated. TriMap, UMAP, and PacMAP use weighted objectives that prioritize local neighborhood preservation. E.g. TriMap constructs triplets primarily from k-NNs, PaCMAP optimizes over three types of pairs (neighbor, mid-near, and far pairs), with dynamically adjusted weights, UMAP optimizes a cross-entropy objective over a nearest-neighbor graph. These methods achieve informative visualizations precisely by giving up on preserving all triplets. This is fully consistent with our result: when $d=2 \ll D$, it is impossible to faithfully preserve all triplet orderings, so any successful visualization method must be highly selective about which structure to retain. Moreover, the well-documented limitations of 2D visualizations support this interpretation [1]. The visualizations are ``informative'' in the sense of revealing local cluster membership, but they do not (and cannot) faithfully represent global distance relationships.
>
> [1]: Wang, et al. (2021). Understanding how dimension reduction tools work: an empirical approach to deciphering t-SNE, UMAP, TriMAP, and PaCMAP for data visualization. JMLR

---

> > ### Author Rebuttal · Reviewer_CDfM · 2026-04-03
> >
> > Dear authors, many thanks for the very useful rebuttal. Incorporating the additional quantitative evaluation would improve the paper a lot, I think. Here are some follow-up questions:
> >
> > **Adding predicted $0.5+\varepsilon$**
> > The match is indeed strikingly good. You chose the constant $c_2=1$. While the predicted accuracies seem a great match, I wonder if that choice actually satisfies the requirements of lemma 2.6.
> >
> > **Real-world data is more structured**
> > Thank you for including these real-world experiments. I think they are very important to highlight applicability and limitations of the paper. As you write, the collapse is much less pronounced for this data, which diminishes the practical applicability of the paper. Nevertheless, I think its merits are still sufficient for publication.
> >
> > **On k-NNs, and methods like TriMap, PacMAP, or UMAP**
> > I think one can equivalently translate a kNN graph to triplet constraints and back. Given a (directed) kNN graph choose as constraints all triplets with anchor and positive sample being points and their k-nearest neighbors and as negative samples all points that are not k-nearest neighbors to the anchor. Conversely, a configuration satisfying all of these constraints has exactly the correct directed kNN graph. Then one could check what share of these constraints are respected in the visualizations (this would be close to metrics like neighbor recall, which are common for evaluating visualizations). I agree that some of these methods like PacMAP do not treat all triplet constraints equally and most of them operate on some symmetrized version of the kNN graph which might not be easily captured by triplet constraints.
> >
> > The second part of my question was purely on kNN graphs. Given the above equivalence between kNN graphs and certain triplet constraints, I wonder if your theory also predicts collapse for kNN graph structure in lower dimensional spaces or whether this type of triplet constraint is "too structured" for your theory to hold.
> >
> >
> > **Remaining theory questions**
> > I am still wondering about the first two questions I had on your theory, i.e., the $-o(1)$ bit and the multi-graph setup.

---

> > > ### Author Response · Authors · 2026-04-07
> > >
> > > We thank the reviewer for the thoughtful follow-up questions/suggestions. We address them below:
> > >
> > > *Q: Adding predicted $0.5+\varepsilon$*
> > >
> > > *Answer:* Regarding the choice of $c_2$: Here, we wanted to emphasize that experimentally the transition from accuracy 50% to 100% follows a square-root curve $Acc = \tfrac12+\sqrt{d/c_2D}$ as predicted in our theory (they agree as they show the same asymptotic behavior, otherwise, no single choice of $c_2$ would allow us to fit the curve so well). Lemma 2.6 requires $c_2$ to be ``small enough,'' and experimentally, we chose the constant $c_2 = 1$ just because it happened to fit the accuracy curve quite well. One way to think about it is that, for certain datasets, the collapse we observed experimentally kicks in already at a reasonable constant $c_2\approx 1$, that does not have to be extremely small, so presumably, the stated Lemma already holds at a wider range of parameters. We will add a figure to clarify this in the revised paper.
> > >
> > >
> > > *Q: Real-world data is more structured*
> > >
> > > *Answer:* As expected and as we observed experimentally, real-world data contains structure that potentially allows for better accuracy at lower dimensions. We believe that the less pronounced collapse here reflects exploitable structure in real embeddings, and is an interesting future research direction on beyond-worst-case contrastive embeddings: essentially, the random model serves as a calibration baseline, and the gap between random and real-data accuracy quantifies the degree of exploitable structure of a dataset (e.g., triplets generated in non-adversarial/non-random way, large-margin assumptions, latent clustered structure etc.).
> > >
> > >
> > > *Q: On k-NNs...*
> > >
> > > *Answer:* We appreciate the reviewer's ideas on k-NNs. Indeed k-NN-triplets offer more structure and we performed the following experiment to explore how they compare to random triplets. Specifically, we ran PaCMAP on the MiniLM+STS-B embeddings ($D = 384, n = 1000$) for $k=10, d = 2$, evaluated the 2D embedding on both cases, and compared it with our main-experiment method. Here, we define ``kNN Triplet Accuracy'' as the reviewer suggested (fraction of satisfied triplets with anchor and positive being k-NNs, and negatives being all other points not k-NNs of the anchor):
> > >
> > > | Method | kNN Trip. Acc. $(k=10)$ | Unif. Trip. Acc.|
> > > |----------|----------|----------|
> > > | PacMAP| 0.8998 $\pm$ 0.0028| 0.6196 $\pm$ 0.0031|
> > > | Free embedding  | 0.8172 $\pm$ 0.0029| 0.6606 $\pm$ 0.0024 |
> > >
> > > The results show a gap between the two cases. While PacMAP (which emphasizes k-NN triplets) performs well w.r.t. k-NN triplets already with $d=2$ dimensions, its accuracy significantly drops when evaluated across all triplets. On the other hand, our free embedding method (used in the main experiment) achieves lower score for k-NN accuracy as it is not specifically tuned to preserve k-NN structure, yet outperforms PacMAP on uniform triplet accuracy.
> > >
> > > We believe extending our results to the k-NN case is a very interesting direction. As is, our Theorem 2.1 applies to uniformly random triplets and not k-NN-derived triplets. Observe that k-NN triplets will not resemble the random set of triplets, as they follow a correlated distribution, depending on the underlying ground-truth geometry. We believe that a similar accuracy collapse may occur for k-NN-triplets though, where the ground-truth embedding only needs $D=c\cdot k$ dimensions to realize all constraints, yet all other representations that use fewer dimensions than $d=k/C$ may suffer from accuracy near 50% (for suitable choices of constants $c,C$).
> > >
> > > We believe this is an interesting open question, and we thank the reviewer for highlighting it.
> > >
> > > *Q: Remaining theory questions*
> > >
> > > *Answer:* For the $-o(1)$ term: this term follows from Lemma 2.5 and Lemma 2.6. By Lemma 2.5, the instance is satisfiable in dimension $D$ with high probability at least $1-o(1)$. On the other hand, by Lemma 2.6 with probability at least $\frac{1}{2}$, no embedding in dimension $d$ satisfies more than $\frac{1}{2}+\epsilon$ of the constraints. We want both of the events to happen, so we will take a union bound: the probability that either of the two events doesn't happen is at most $\frac{1}{2}+o(1)$, thus both of the events happen with probability at least $1-(\frac{1}{2}+o(1))=\frac{1}{2}-o(1)$.
> > >
> > >
> > > Regarding multigraphs: We emphasize that at the density we are studying, the graph is simple (every edge appears at most once) with high probability $1-o(1)$, therefore for all intents and purposes the reader can have a simple graph in mind. However to answer your question, if an edge appears multiple times then it is counted multiple times towards the accuracy calculation.
> > >
> > > Regarding directed cycles, we only care that a directed cycle does not exist in the graph (if one exists the  multiplicity of the edges is irrelevant as the instance becomes unrealizable), so when counting directed cycles, we don't double count cycles if edges appear more than once.

---

### Official Review · Reviewer_iVZF · 2026-03-13

**Soundness:** 4
**Presentation:** 4
**Significance:** 4
**Originality:** 4
**Overall Recommendation:** 6
**Confidence:** 5

**Summary:**

This work studies the problem of learning embedding of n points into a d dimensional space subject to satisfying a set of triplet constraints. There is an information-theoretic limit result that says there exists a ``perverse'' set of triplet constraints that is embeddable in D but for d bounded away from D, then it is essentially impossible to do better than random guessing with a d-dimensional embedding.
A second result is the NP-hardness (assuming unique games conjecture) of distinguishing between a set of triplets that are approximately realizable versus a set of triplets that are perverse.

**Compliance With Llm Reviewing Policy:**

Affirmed.

**Key Questions For Authors:**

How close to 1 can the c(epsilon) be in Part 2 of Theorem 1.3?

Do we (i.e., the literature) have any insight on generalizability when using, say, the hinge triplet lossas on line 407 left column?

**Limitations:**

yes

**Strengths And Weaknesses:**

I find the result to be an impressive piece of theory for contrastive learning. The proof strategy is also quite elegant and leverages a neat fact about random graphs that the author prove, and also some creative use of recent result on VC theory for contrastive learning with triplets.

I find the work to be very strong and a substantive contribution. There are areas that the papers do not cover, such as the flip-side (when is learning easy, theoretically) of the story. However, I do not believe these are weaknesses as this work is of theoretical nature. I will go into them in the questions section.

---

> ### Author Rebuttal · Authors · 2026-03-31
>
> We thank the reviewer for the very positive assessment (as they write ``very strong and a substantive contribution''), for carefully checking the mathematics, and for recognizing the elegance of our proof strategy. Below we address their questions:
>
> ____
>
> *Question: How close to 1 can the $c(ε)$ be in Part 2 of Theorem 1.3?*
>
> Answer: This is an interesting question. The ideal/strongest result here, that would highlight the importance of choosing the right $d$ would be if we could have a construction of a dataset, that is realizable with $D$ dimensions (same as part 1 of our Theorem 1.3), yet *every* embedding of dimension $d=D-1$ already fails to achieve accuracy better than the random baseline of $1/2$. However, we are not aware of such constructions that allow for such sharp tradeoffs, and in fact, we believe that such strong statements do not hold for the task of contrastive learning from triplets, studied in our work. We believe it is an interesting direction to find constructions where $c(\epsilon)$ is very close to 1.
>
> In our result, the stated bound comes from using the universal learning theory result for uniform convergence (combining with the Hoeffding bound), and for our approach this bound is tight, i.e., having $c(ε)=ε^2$ is the best that we can obtain. Perhaps there could be some other more clever construction, that yields better dependence on $d$ vs $D$. We note that experimentally, we verified this dependence for our construction, as the predicted accuracy $Acc = 1/2 + \sqrt{d/D}$ nearly matches the observed behavior in several contexts.
>
> ____
> *Question: Do we (i.e., the literature) have any insight on generalizability when using, say, the hinge triplet loss as on line 407 left column?*
>
> Answer: We interpret the question as whether or not having low triplet loss can be translated to having low $0/1$ loss for unseen triplets. We are not aware of such clean statements, and perhaps it is interesting to investigate how the margin used in the hinge triplet loss can affect such generalization behavior. More on the practical side, the hinge triplet loss is popular as it was first used for face recognition tasks with great success (FaceNet paper [1]). A close variant of the hinge triplet loss, is the soft-max version studied in Saunshi et al. [2], where they show that optimizing for the soft-max version will yield success in downstream tasks, under certain latent structure assumptions on the datasets. For more on the different optimization methods and their performance for such contrastive learning tasks (also closely related to ordinal embeddings), please see the systematic evaluation in the survey [3].
>
>
> [1] Schroff, Florian, Dmitry Kalenichenko, and James Philbin. "Facenet: A unified embedding for face recognition and clustering." Proceedings of the IEEE conference on computer vision and pattern recognition. 2015.
>
> [2] Saunshi, N., Plevrakis, O., Arora, S., Khodak, M., & Khandeparkar, H. (2019, May). A theoretical analysis of contrastive unsupervised representation learning. In International conference on machine learning (pp. 5628-5637). PMLR.
>
> [3] Vankadara, Leena Chennuru, et al. "Insights into ordinal embedding algorithms: A systematic evaluation." Journal of Machine Learning Research 24.191 (2023): 1-83.

---

> > ### Author Rebuttal · Reviewer_iVZF · 2026-04-07
> >
> > Thank you for the reply.

---

### Official Review · Reviewer_7oEP · 2026-03-13

**Soundness:** 3
**Presentation:** 4
**Significance:** 3
**Originality:** 3
**Overall Recommendation:** 5
**Confidence:** 3

**Summary:**

The authors prove that, in triplet learning settings, there is a necessary collapse in accuracy when using $cD$ dimensions to embed data which exists "naturally" in $D$ dimensions. They provide experiments which verify the intuition presented in the theorems.

**Compliance With Llm Reviewing Policy:**

Affirmed.

**Key Questions For Authors:**

My questions are raised in the above section on weaknesses. I am curious about extensions of the work to "real-world" data settings where $D$ isn't known a-priori, as well as to how the inapproximability might guide intuition about when the inapproximability applies to real-world data.

**Limitations:**

Yes.

**Strengths And Weaknesses:**

## Strengths

The paper is overall excellent. It is a joy to read. The ideas are laid out extremely clearly and, although I am not fully immersed in the literature of TCS-flavored theory towards contrastive learning ideas, I found the ideas natural. Here are a few points which are particularly strong:
- The proof that the collapse happens at a dimensionality which scales with the ground truth dimensionality is interesting (and slightly surprising!)
- The inapproximability is a nice complement to the main result.
- The experiments are the natural test of the result and I appreciate their presence as it helps with the intuition.

## Weaknesses

I think the paper's primary weakness is that it is not clear how one might *apply* these ideas to the practical context. For one thing, the paper assumes that the ground-truth dimensionality $D$ is known. However, this is a very strong assumption! In the context of standard image-based contrastive learning, this assumption is immediately broken, for example. Indeed, it is not even clear to me what is meant by ground-truth dimensionality in the context of, for example, image-based contrastive learning. Is it the dimensionality of the vectorized image or is it the dimension of some "manifold" (for lack of a better word) that the data lives on?

Similarly, it is not clear to me how to think through the implications of the inapproximability. Indeed, we find over and over again that models *are* able to learn the approximate triples. So does this imply that, while it is hard to distinguish YES and NO instances, a majority of realistic settings are in fact YES instances? Or am I missing something here?

---

> ### Author Rebuttal · Authors · 2026-03-31
>
> We thank the reviewer for the very positive evaluation and the constructive comments. Below we answer their questions:
> _______________
> *Question: ``Indeed, it is not even clear to me what is meant by ground-truth dimensionality in the context of,
> for example, image-based contrastive learning. Is it the dimensionality of the vectorized image or is it the dimension
> of some ”manifold” (for lack of a better word) that the data lives on?''*
>
> Answer: This is an important conceptual question. We clarify that capital $D$ in our framework is the minimum dimension at which
> the triplet constraints of interest are perfectly realizable, i.e., embeddable so that all given triplets are in agreement with the embedding. In other words, it is the smallest dimension needed to preserve the ordinal structure (not length of distances) in the dataset. Our main result is information-theoretic, so we don’t assume that $D$ is known (since we do not give an algorithm that needs to use $D$), and our main theorem works for every dimension $D$ above the stated threshold. In the context of images, $D$ is not be the vectorized image dimension which could be very large, but rather the dimension of an embedding that preserves all triplet comparisons; depending on the structure of the data this can be much smaller. We note that if real-world data are not realizable (e.g., due to noisy inputs), then a natural notion for the ground-truth dimenion $D$ is the smallest dimension so that the triplet accuracy is above a satisfactory user-defined threshold (say 98%). One crisp way of interpreting our findings, is that a careful choice for the embedding space $d$ in comparison to $D$ can go a long way, as our result highlights that even a constant approximation to $D$ may not yield accuracy better than random (that is already achievable on the line with $d=1$).
>
> _______________
>
> *Question: ``It is not clear to me how to think through the implications of the inapproximability.''*
>
> Answer: Our optimal inapproximability result is a worst-case result (under UGC): it says that there exist hard instances where no polynomial-time algorithm can beat the trivial $1/2 + ϵ$ baseline, even if a near-perfect solution exists. This highlights the limitations of algorithmic results achievable for the contrastive learning problem, in that there cannot be a universal algorithm (for images, text, video etc.) that runs in polynomial-time and achieves better accuracy than $1/2$. Of course, this does not contradict the practical success on specific contrastive learning tasks for real-world data. Just as 3SAT is NP-hard in the worst case, yet practical SAT solvers handle structured instances efficiently, our hardness result identifies the computational frontier without excluding practical success on structured inputs. Our result just excludes a general result that would work for all instances; however, under specific assumptions, one may in principle get better results, depending on the application at hand. Indeed, addressing this gap between theory and practice, and formulating well-crafted strucutral assumptions on real-world data depending on the task (different assumptions needed for images, or text, or other types of applications) is currently a fruitful direction in the literature of beyond-the-worst-case analysis of algorithms and machine learning (see Roughgarden's book [1] on the topic).
>
> [1] Roughgarden, Tim, ``Beyond the worst-case analysis of algorithms,'' Cambridge University Press, 2021.
>
> _______________

---

> > ### Author Rebuttal · Reviewer_7oEP · 2026-04-03
> >
> > I appreciate the authors' response. I have no other questions at this time.

---

### Decision · Program_Chairs · 2026-04-30

**Decision:**

Accept (spotlight)

**Comment:**

The paper develops mathematical theory to study accuracy of $d$-dimensional embeddings under triplet constraints. The authors prove that accuracy can strongly decrease close to chance level when $d=cD$, where $c$ is a constant and $D$ is ground-truth data dimensionality. The authors conduct several experiments to provide empirical support to their theorems.

All four reviewers liked the paper and found it "very strong", "excellent", and "a joy to read". They raised some questions about the empirical section, and the authors provided additional evidence that the experiments do match the theory. My recommendation is strong accept.

What follows are some additional comments that I had when reading the paper (note that I only read Sections 1 and 4):

* I did not like the "collapse" framing that the authors use ("collapse" in the title", "sudden collapse" in the abstract, etc.). The way title/abstract are written, I expected to see a sharp drop in accuracy at some specific value of $d$. The abstract says that unless $d$ is close to $D$, "accuracy undergoes a sudden collapse". But in fact I do not see any sudden collapse in Figures 1 and 2! I see a monotonic decrease from 100% to 50% as $d$ decreases from $D$ to 1. That the accuracy should decrease, is obvious a priori. I would suggest to replace "accuracy collapse" with "accuracy decrease" throughout the paper, including in particular title/abstract.

* The sentence in the abstract that states the main result (line 26) seems to be not sufficiently precise. What the paper proves, is that for any given accuracy $0.5 + \epsilon$, there is some constant $c$ such that etc. But the abstract says "every embedding violates half of the triplets" which is confusing. Not half, but half + eps! And the constant $c$ depends on the value of epsilon! Please rewrite this sentence so that it makes sense and so that it is clear what the constant $c$ depends on (epsilon? number of triplets? dataset size? etc.).

* Theorem 1.3 states the main result in the Introduction and says that $c$ is a function of $\epsilon$. But the Introduction does not say that $c\approx \epsilon^2$. So after skipping to Section 4, I was confused to read that the theory predicts $c=\epsilon^2$ (see paragraph called "Plots"). Please add something to the Introduction about this dependency.

* As several reviewers also noticed, it would be great to add $0.5 + \sqrt{d/D}$ line to Figures 1 and 2, if this is the prediction of your theory.

* I noticed that in the experiments, constrained embeddings work much worse than unconstrained embeddings and never reach accuracy 1. In the bottom panel of Figure 1, the red line for D=1024 plateaus at 0.8, the blue line for D=128 plateaus at 0.9, whereas in the upper panel both reach 1.0. The same in Figure 2. Why is that? Please add some comments and explicit discussion of it to the text. To me it seems that your optimization procedure somehow fails with constrained embeddings. If this is also your interpretation, then maybe remove the constrained results to the appendix? Unless you manage to fix/improve the optimization...